

# Transcriptome analysis provides a blueprint of coral egg and sperm functions

Julia Van Etten[1], Alexander Shumaker[2], Tali Mass[3], Hollie M. Putnam[4] and Debashish Bhattacharya[5]

[1] Graduate Program in Ecology and Evolution, Rutgers, The State University of New Jersey, New Brunswick, NJ, United States of America
[2] Microbial Biology Graduate Program, Rutgers, The State University of New Jersey, New Brunswick, NJ, United States of America
[3] Department of Marine Biology, University of Haifa, Haifa, Israel
[4] Department of Biological Sciences, University of Rhode Island, Kingston, RI, United States of America
[5] Department of Biochemistry and Microbiology, Rutgers, The State University of New Jersey, New Brunswick, NJ, United States of America

Corresponding author
Debashish Bhattacharya,
debash.bhattacharya@gmail.com,
d.bhattacharya@rutgers.edu

## ABSTRACT

**Background**. Reproductive biology and the evolutionary constraints acting on dispersal stages are poorly understood in many stony coral species. A key piece of missing information is egg and sperm gene expression. This is critical for broadcast spawning corals, such as our model, the Hawaiian species *Montipora capitata*, because eggs and sperm are exposed to environmental stressors during dispersal. Furthermore, parental effects such as transcriptome investment may provide a means for cross- or trans-generational plasticity and be apparent in egg and sperm transcriptome data.
**Methods**. Here, we analyzed *M. capitata* egg and sperm transcriptomic data to address three questions: (1) Which pathways and functions are actively transcribed in these gametes? (2) How does sperm and egg gene expression differ from adult tissues? (3) Does gene expression differ between these gametes?
**Results**. We show that egg and sperm display surprisingly similar levels of gene expression and overlapping functional enrichment patterns. These results may reflect similar environmental constraints faced by these motile gametes. We find significant differences in differential expression of egg vs. adult and sperm vs. adult RNA-seq data, in contrast to very few examples of differential expression when comparing egg vs. sperm transcriptomes. Lastly, using gene ontology and KEGG orthology data we show that both egg and sperm have markedly repressed transcription and translation machinery compared to the adult, suggesting a dependence on parental transcripts. We speculate that cell motility and calcium ion binding genes may be involved in gamete to gamete recognition in the water column and thus, fertilization.

## INTRODUCTION

Reef-building corals and their photosynthetic dinoflagellate endosymbionts form the structural foundation of complex ecosystems, supporting approximately 25% of marine biodiversity and protecting coastlines from damaging wave energy (*Hughes et al., 2017*). The rice coral *Montipora capitata* is a dominant reef-builder in the Hawaiian Archipelago that is of interest because of its demonstrated resilience throughout ocean warming events that can lead to loss of algal symbionts (termed ''bleaching'') and mortality (*Grottoli, Rodrigues & Palardy, 2006*). *M. capitata* has a relatively large genome (ca. 886 Mbp in size) due primarily to high repeat and transposable element content. This genome expansion likely resulted from genetic drift due to small effective population size in the isolated Hawaiian island chain (*Shumaker et al., 2019*). These features, along with its relatively high tolerance to heat stress and ocean acidification (*Gibbin et al., 2015*), make *M. capitata* an important model, when compared to more sensitive coral species, for studying the emergence of locally adaptive traits and physiological responses of corals to environmental change.

*M. capitata* is a broadcast-spawning hermaphrodite that annually releases bundles of sperm and eggs into the water column at roughly three summer intervals, according to the lunar cycle. Gametes undergo sexual fusion and produce larvae that settle, metamorphose, and develop into the meta-organisms that ultimately build reefs (*Padilla-Gamiño et al., 2011*; *Padilla-Gamiño & Gates, 2012*; *Mass et al., 2016*). Broadcast-spawning is the most common form of reproduction within Scleractinia (stony corals) and is highly conserved within this order (*Baird et al., 2009*; *Padilla-Gamiño et al., 2011*). Unlike brooding corals, *M. capitata* gametes experience direct and prolonged exposure to the marine environment (within bundles, and thereafter upon release at the sea surface) during the several days of development and ∼weeks of pelagic larval duration (*Concepcion, Baums & Toonen, 2014*). Gamete survival in the water column is a period of a few hours during which they must abide environmental challenges such as predation, infection by microbial pathogens, and fluctuations in temperature and pH, before taking part in the fertilization process (*Baird et al., 2009*) and larval recruitment. Despite the significance of gamete survival in the marine environment and its effect on overall recruitment potential in broadcast-spawning corals, few studies have focused primarily on egg and sperm cells and only one recent study has investigated gamete transcriptomics in the context of gametogenesis (*Chiu et al., 2020*). Moreover, investigation of the potential impacts of bleaching on reproduction has not shown significant differences in egg quality or spawning potential between bleached and non-bleached *M. capitata* (*Cox, 2007*), despite the potential for parental investment in egg algal symbiont populations through vertical transmission. This reproductive resilience may be explained in part by an increase in adult heterotrophy, offsetting energetic losses due to bleaching (*Grottoli, Rodrigues & Palardy, 2006*), or the timing of gametogenesis relative to the thermal stress. However, bleaching events have been shown to negatively impact the degree of spawning for years after their occurrence, even in corals that did not show visible bleaching during the initial event (*Levitan et al., 2014*), as well as some gamete characteristics including reduced egg volume, longer time to first cleavage, and sperm

motility (*Hagedorn et al., 2016*). Sperm play an equally important role in determining spawning success, yet little is known about sperm gene expression. Sperm DNA has been used to generate genome assemblies due to its high seasonal abundance and symbiont-free state, therefore data are readily obtainable for sperm-specific studies (e.g., *Putnam et al., 2017*; *Shumaker et al., 2019*).

Here, we analyzed egg and sperm RNA-seq data generated from *M. capitata* colonies located on fringing reefs near the Hawai'i Institute of Marine Biology (HIMB) in O'ahu (for details on related analyses, see *Putnam et al., 2017*). Six individual adult colonies were sampled to obtain three individual egg RNA-seq libraries and three individual sperm RNA-seq libraries (both from ambient conditions) and were sequenced on the Illumina platform using the Illumina TruSeq RNA Library Preparation Kit v2. The egg data are publicly available under NCBI BioProject PRJNA616341 (SAMN14486762, SAMN14486763, SAMN14486764) and the sperm data are publicly available under NCBI BioProject PRJNA339779. We determined which genes are expressed in coral egg and sperm cells and studied their putative functions. We compared cDNA data from eggs and sperm to that of RNA isolated from tissues of three individual un-stressed adults generated in a previous project (*Shumaker et al., 2019*) to determine the degree of differential gene expression (DEG) between gametic and adult tissues. We then compared gene expression of egg and sperm to identify functions that are shared or distinguish these gametes. Finally, we used the expression data to highlight pathways represented by the most differentially expressed genes in egg and sperm and compared gamete-specific functions that account for various known physiological processes in *M. capitata*.

## MATERIALS & METHODS

### Sperm sample collection and sequencing library preparation

*M. capitata* sperm were collected from the fringing reefs on the west side of Moku o Lo'e and RNA extracted as described in *Putnam et al. (2017)*. Three individual RNA-seq libraries were generated using 200 ng of the total RNA from each sample using the Illumina TruSeq RNA Library Preparation Kit v2. The libraries were combined in equimolar concentrations and run on a single Illumina MiSeq flowcell using the Illumina MiSeq Reagent Kit v3 (150 cycles, single-end). The genomic and transcriptomic data created for this project are available under NCBI BioProject PRJNA339779 and the three libraries were pooled together and can be found in BioSample: SAMN05607941; Sample name: Mcap_nonBleach_RNA; SRA: SRS1632867.

### Transcriptomic and genomic data analysis of sperm

After trimming for quality (parameters shown in Fig. S1) using the CLC Genomics Workbench 8.5.1 (Qiagen, Hilden, Germany), the combined sperm RNA-seq data yielded 29,815,942 high quality reads for assembly. The *M. capitata* genome (*Shumaker et al., 2019*) and structural annotations used for the mappings are available at http://cyanophora.rutgers.edu/montipora/. The libraries were individually mapped to the reference genome using CLC Genomics Workbench (count data can be found in Table S1).

Only "unique exon read" counts (i.e., the number of reads that match uniquely to exons, including across exon-exon junctions) were used for downstream analyses.

## Egg sample collection

In the June 2018 (6/13/18) spawning period, egg-sperm bundles were collected from ambient conditions in the field on the fringing reefs at the HIMB under Special Activity Permit 2018-50 from the Hawai'i Department of Aquatic Resources. Bundles were brought back to the lab and buoyant eggs were separated from the dense sperm after the bundles broke up. The eggs were subsequently rinsed 3 times with 0.2 µm filtered seawater and then following the removal of the water, replicate tubes of eggs were snap frozen in liquid nitrogen and stored at −80 °C.

## Egg RNA extraction and sequencing library preparation

RNA was extracted using a dual DNA/RNA extraction method (Quick-DNA/RNA Miniprep Plus Kit, Cat# D7003). A total of 300µl of Zymo DNA/RNA shield, 30 µl of PK digestion buffer was added to each sample tube, followed by the addition of 5µl Proteinase K. Next, samples were vortexed, spun down, and placed in a Thermomixer for 1 h at 55 °C, shaking at 1,100 rpm. After heating, samples were and centrifuged at 2,200 rcf for one minute to separate any remaining solids. The supernatant was transferred to a new microfuge tube and equal volume of DNA/RNA Lysis Buffer was added and mixed. All liquid was transferred to a new Spin-Away filter column with collection tube and spun at 16,000 rcf for 30 s. The flow through containing the RNA fraction was added to a 1.5 mL tube with an equal volume of 100% EtOH was and tubes were vortexed and spun down. A total of 700 µl of the resulting mixture was added to the RNA spin columns where it was centrifuged at 16,000 rcf (g) for 30 s and the flow through (Zymo kit waste) was subsequently discarded. This step was repeated and then 400 µl DNA/ RNA Wash Buffer was gently added to each RNA column. The samples were centrifuged at 16,000 rcf (g) for 30 s and the flow through (Zymo kit waste) was discarded. Next, 80 µl DNase I treatment master mix (75 µl DNA Digestion buffer × # of samples, 5 µl DNase I × # of samples) was added directly to the filter of the RNA columns and incubated at room temp for 15 min. Then, 400 µl of the DNA/ RNA Prep Buffer was gently added to each column and the mixture was centrifuged at 16,000 rcf (g) for 30 s and the flow through (Zymo kit waste) was discarded. The columns containing the bound RNA were then transferred to new 1.5 mL microcentrifuge tubes and 50 µl of warmed DNase/RNase free water was added to each RNA column by dripping slowly directly on the filter. The samples were incubated at room temperature for 5 min and centrifuged at 16,000 rcf (g) for 30 s. This step was repeated to obtain a final elution volume of 100 µl, and tubes were stored at −80 °C. The cDNA libraries were prepared and sequenced by Genewiz using standard Illumina strand-specific RNA-seq preparation with poly-A selection and then sequenced with the HiSeq instrument using $2 \times 150$ bp reagents with ∼15M raw paired-end reads per sample.

## Transcriptomic and genomic data analysis of eggs

After trimming for quality (parameters shown in Fig. S1) using the CLC Genomics Workbench 8.5.1 (Qiagen, Hilden, Germany), the combined egg RNA-seq data

yielded 125,054,213 high quality paired reads for assembly. The *M. capitata* genome (*Shumaker et al., 2019*) and structural annotations used for the mappings are available at http://cyanophora.rutgers.edu/montipora/. The libraries were individually mapped to the reference genome using CLC Genomics Workbench 8.5.1 (QIAGEN, Aarhus Denmark) (count data can be found in Table S2). Only "unique exon read" counts were used for downstream analyses.

## Functional analysis
### Read count normalization
Read counts across the three egg and three sperm libraries, as well as three adult ambient condition RNA-seq runs from *Shumaker et al. (2019)* used in DEG analysis, were normalized to transcripts per million (TPM). The number of genes expressed at TPM thresholds between 0 and 200 (in increasing increments of 10) were tabulated and used to determine the proportion of the gene inventory expressed at each threshold. These values were converted to percent of total genes in the genome, and that percentage was plotted on the $y$-axis vs. the incremental thresholds on the $x$-axis. Based on the graphs for each of the three datasets (egg, sperm, adult), there is a steep drop off in the percentage of genes around 100 TPM in both the egg and sperm data, where >100 TPM represents 12.05% (7,620/ 63,227) and 12.31% (7,783/ 63,227) of the total genes, respectively. In the adult read counts, ~12% of all genes corresponds to 60 TPM, however, because this group had many more expressed genes (TPM>0, 58.69%) compared to the egg and sperm (31.98% and 30.27% respectively), the drop-off in percentages of genes above the TPM threshold increments occurs very early in this dataset. Because the adult data are not relevant to this particular part of the analysis, we chose 100 TPM to be a reasonable threshold for moving forward with the egg and sperm data because although arbitrary, this threshold reflects consistency in proportion of the genes represented across these datasets. Normalized read counts and accompanying graphs are provided in Table S2.

### Gene expression in egg and sperm based on gene ontology
To determine which biological, cellular, and molecular functions are most prevalent across the combined egg and combined sperm RNA-seq data, we used the 100 TPM threshold to generate a list of "expressed" genes in both the egg and sperm which consisted of 7,620 and 7,783 genes respectively. The Blast2GO software was used to map GO terms to the *M. capitata* gene inventory and to test for enrichment among the sets of "expressed" genes using Fisher's Exact Test (*Götz et al., 2008*), using the gene inventory as the reference set.

## Using GO terms to organize KEGG pathway data
These sets of "expressed genes" were assigned KEGG Orthology (KO) terms using the KofamScan software (*Aramaki et al., 2019*), yielding 4,078 genes in the "egg" set with at least one KO term and 3,146 genes in the "sperm" set with at least one KO term. Each set of KOs (egg, sperm) was mapped onto well-studied biological pathways using the "Reconstruct Pathway" tool (*Kanehisa, 2017*). Next, all KEGG pathways associated with each KO (excluding human-specific pathways) were retrieved and cross-referenced with GO terms of the category "cellular component". This allowed for assignment of each KO
term and its associated pathway(s) to a putative location within the cell. The exhaustive list of all KEGG pathway-GO pairs is included in Tables S4 and S6 and the most common terms relevant to each cellular component were selected as the functional pathways to display in our analysis. This analysis was done to make an overall assessment of which metabolic processes are most frequently expressed in egg and sperm cells based on the normalized read counts for RNA-seq transcript data, and to identify if there are any major differences in overall functionality between each gamete type that can be further investigated.

### Differential gene expression analysis

DEG analysis was done in RStudio 3.5.3 (*RStudio Team, 2018*) using the DESeq2 package (*Love, Huber & Anders, 2014*) using the raw counts as input (parameters shown in Fig. S2). The count matrices and column data used as input can be found in Tables S8, S9, and S12. Differentially expressed genes were identified in contrasts between egg and adult samples and sperm and adult samples, as well as between egg and sperm samples. Egg replicates corresponding to individual RNA-seq runs are labeled E1, E2, E3, sperm samples are labeled Ub2, Ub3, Ub4, and ambient control treatment adult samples from *Shumaker et al. (2019)* are labeled W1, W5, and W7 in all related tables and figures.

## RESULTS

### Functional analysis

Fisher's exact test revealed similar functional enrichments among the "expressed" gene sets (TPM > 100) in both gamete types. Among the Biological Process and Molecular Function GO terms enriched in the "expressed" genes in the egg samples, the top ten are 'ATP binding', 'GTP binding', 'GTPase activity', 'structural constituent of ribosome', 'chromatin binding', 'microtubule motor activity', 'negative regulation of transcription by RNA polymerase II', 'protein polyubiquitination', 'translation initiation factor activity', and 'ubiquitin ligase activity'. The top ten enriched GO terms for sperm "expressed" vs. all genes are 'ATP binding', 'GTP binding', 'GTPase activity', 'structural constituent of ribosome', 'chromatin binding', 'microtubule binding', 'endonuclease activity', 'microtubule motor activity', 'negative regulation of transcription by RNA polymerase II', and 'protein polyubiquitylation'. The complete list of the GO terms for each of these enrichment profiles and their dataset of origin are presented in Table 1. These GO lists, their prevalence in the *M. capitata* genome, and degree of enrichment are shown in Fig. S3. All GO term enrichments beyond the top 30 shown here were minor and not reported.

### Using GO terms to organize KEGG pathway data

After indexing KO terms by their corresponding Cellular Component GO terms, we determined how much of each transcriptome (TPM of all genes assigned to each GO term divided by total TPM of all genes in each "expressed" dataset) is associated with each cellular component and found this metric to be very similar between the egg and sperm datasets (see Fig. 1). To then determine the underlying functional pathways associated with these locations within the cell, the top KEGG pathways for selected parts of the cell were identified, excluding those that are human/ mammal-specific or disease-related. Each KO
**Table 1  Enriched egg and sperm GO terms.** List of enriched GO terms found in the egg, sperm, and egg and sperm "expressed" gene sets.

| Egg "expressed" enriched GO terms | Enriched GO terms shared by egg and sperm "expressed" gene sets | Sperm "expressed" enriched GO terms |
|---|---|---|
| Guanyl-nucleotide exchange factor activity | ATP binding | Microtubule binding |
| Phosphoric diester hydrolase activity | GTP binding | Endonuclease activity |
| Meiotic cell cycle | GTPase activity | Ubiquitin protein ligase binding |
| Cysteine-type endopeptidase activity | Structural constituent of ribosome | Translational elongation |
| Protein processing | Chromatin binding | SNARE binding |
| DNA-templated transcription, elongation | Microtubule motor activity | Protein secretion |
| | Protein polyubiquitination | Protein localization to plasma membrane |
| | Translation initiation factor activity | RNA helicase activity |
| | Ubiquitin protein ligase activity | RNA methyltransferase activity |
| | Negative regulation of transcription by RNA polymerase II | |
| | Thiol-dependent ubiquitin-specific protease activity | |
| | Rab protein signal transduction | |
| | Unfolded protein binding | |
| | Single-stranded DNA binding | |
| | Actin filament binding | |
| | Magnesium ion binding | |
| | Protein heterodimerization activity | |
| | GTPase activator activity | |
| | Rab GTPase binding | |
| | Peptidyl-serine phosphorylation | |
| | RNA helicase activity | |
| | NAD binding | |

term was counted for each part of the cell and the most frequent terms were included in Fig. 2. The complete list of all of the raw data used in this analysis can be found in Tables S5 and S7. These results further support the Biological Process and Molecular Function GO data shown above by confirming the similarity between the egg and sperm with respect to which pathways are most prominently expressed and where they are active within the cell. The functional implications of these differences will be discussed below.

## Differential gene expression analysis
### Gamete vs. adult

From a broader perspective, and perhaps surprisingly, *M. capitata* egg and sperm appear to have similar patterns of gene expression and share functional enrichment patterns. However, in order to gain a more in-depth understanding of how each of these cells function, how those functions compare with gene expression in adult *M. capitata* tissue, and how egg and sperm gene expression is related to each other, it is crucial to statistically test for differential gene expression. As expected, given the divergent different tissue types analyzed, principle components analysis (PCA) shows strong differentiation in gene expression patterns between gametic and adult libraries (Fig. S4).

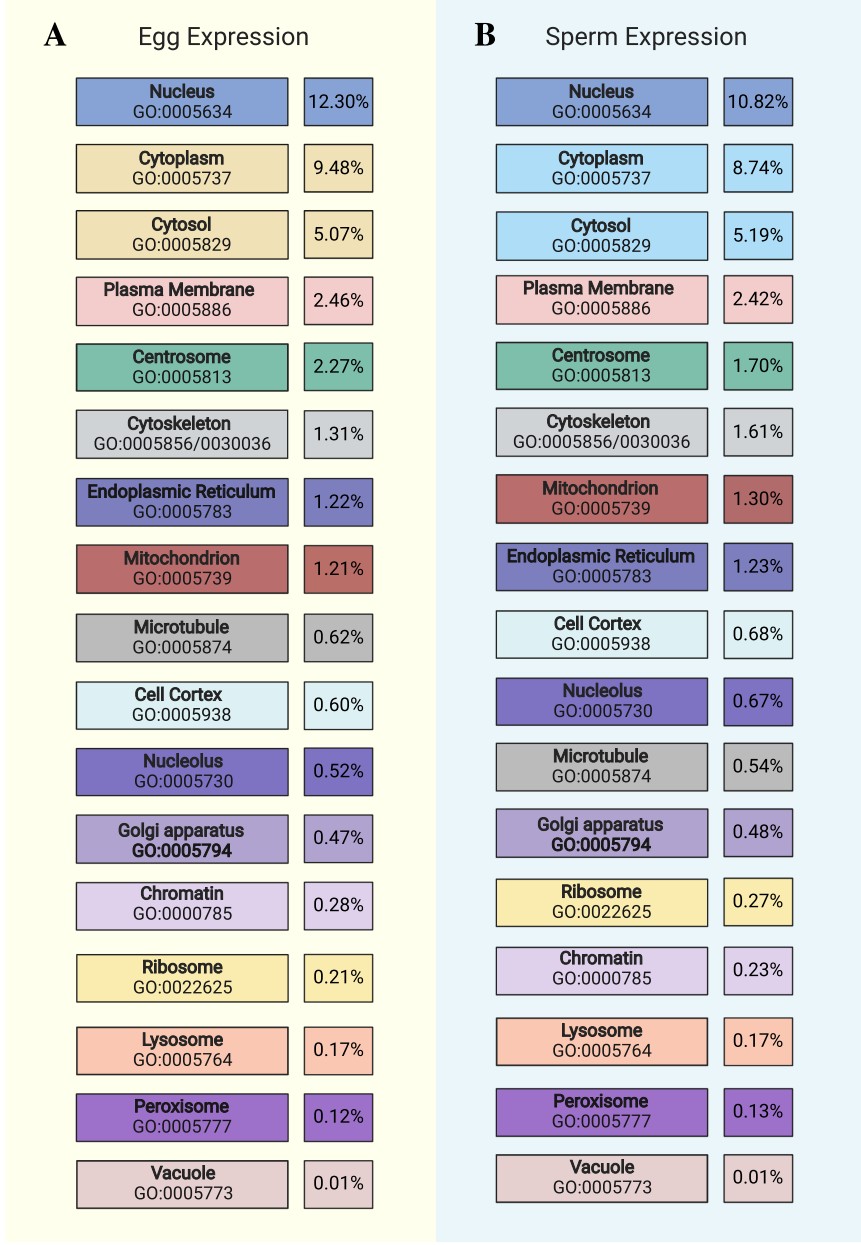

**Figure 1  Distribution of gene expression in egg and sperm.** Chart showing cellular locations and accompanying GO terms (color-coded to Fig. 2) comparing (A) egg and (B) sperm data with respect to the proportion of the respective "expressed" datasets (TPM > 100) that are ascribed to each cellular component. This chart is not an exhaustive list of all cellular components (see Tables S4 and S6 for the full list) but highlights prominent features. Image created with https://biorender.com/.

More specifically, with respect to the DEG analysis of egg vs. adult RNA-seq data, DESeq2 identified 13,890 transcripts that were significantly differentially expressed (defined in this study as FDR-adjusted $p$-value < 0.05), of which 4,487 were up-regulated and 9,403 down-regulated. With respect to the differential expression analysis of sperm vs. adult RNA-seq

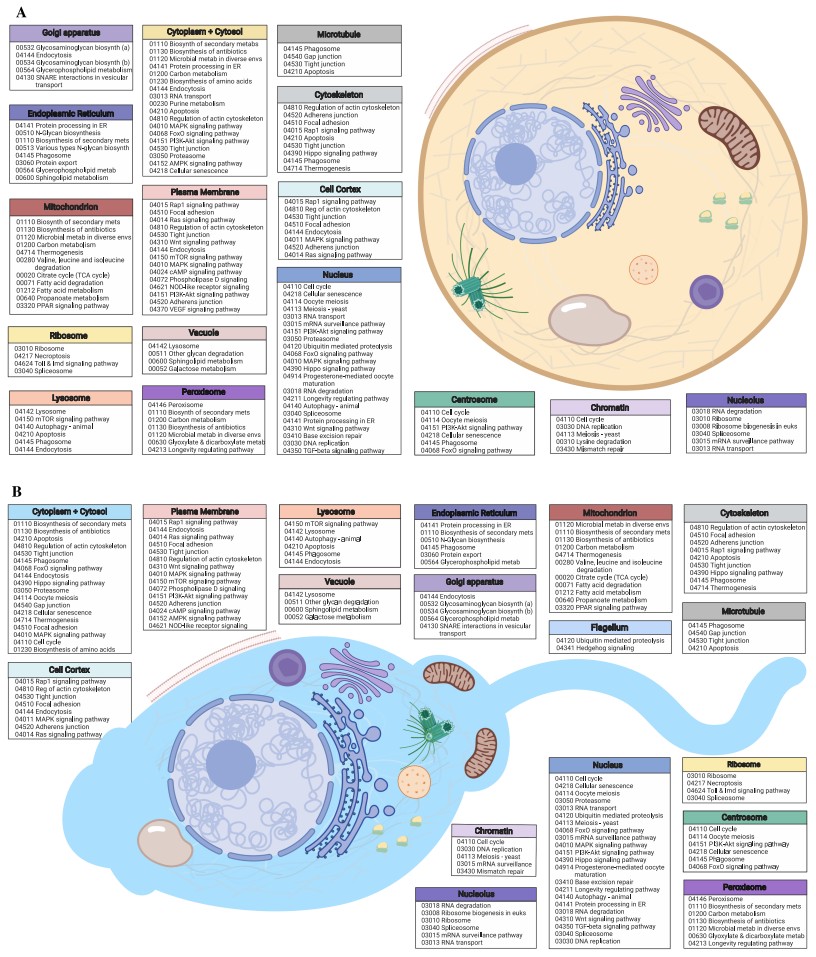

**Figure 2** **Egg and sperm overall functions.** (A) Schematic image of an *M. capitata* egg cell depicting cellular structures and organelles associated with GO terms and their accompanying KEGG pathways determined from KO terms. (B) Schematic image of an *M. capitata* sperm cell depicting cellular structures and organelles associated with GO terms and their accompanying KEGG pathways determined from KO terms. Image created with https://biorender.com/.

data, DESeq2 identified 9,717 transcripts that were significantly differentially expressed (defined in this study as FDR-adjusted $p$-value < 0.05), of which 2,985 were up-regulated and 6,732 down-regulated. Log2-fold change estimates (L2FC), all FDR-adjusted $p$-values, putative annotations, blastx hits and percent identity (PID), KO terms, and GO terms are provided for all of these genes the Tables S10 and S11. The top ten DEGs that had BLAST hits for egg up- and down-regulated and sperm up- and down-regulated are shown in Tables S3 and S4 with all of their differential expression and BLAST statistics and the full lists of DEGS and accompanying annotations are in Table S3. It is difficult to use these data to gain specific insights because many of the DEGs lack BLAST hits, and those that do, most are to predicted, hypothetical, or uncharacterized proteins in recently sequenced corals. This trend is illustrated in Fig. S5 that provides a snapshot of the degree of similarity

between the egg and sperm DEG lists as well as the limitations associated with DEGs that have poor annotation.

To understand some of the basic functions of the egg and sperm transcriptomes, four sets of DEGs were used as the test sets for separate GO-enrichment analyses (Fisher's Exact Test) against the reference set of all genes in the *M. capitata* genome (Blast2GO OmicsBox 1.1.164; *Götz et al., 2008*): (1) egg up-regulated genes (L2FC >1.5; 3,645 DEGs), (2) egg down-regulated genes (L2FC <−1.5; 8,760 DEGs), (3) sperm up-regulated genes (L2FC >1.5; 2,795 genes), and (4) sperm down-regulated genes (L2FC <−1.5; 6,606). The top enriched GO terms for egg up-regulated genes are "ATP binding", "phosphatase activity", "positive regulation of biosynthetic process", "protein serine/threonine kinase activity", and "GTP binding". The top enriched GO terms for egg down-regulated genes are "signaling receptor activity", "regulation of transcription by RNA polymerase II", "lipid metabolic process", "organonitrogen compound catabolic process", and "regulation of signaling". The top enriched GO terms for sperm up-regulated genes are "purine nucleotide binding", "purine ribonucleoside triphosphate binding", "enzyme binding", "positive regulation of RNA metabolic process", and "regulation of localization". The top enriched GO terms for sperm down-regulated genes are "positive regulation of cellular metabolic process", "regulation of transcription by RNA polymerase II", "carboxylic acid metabolic process", "regulation of cellular protein metabolic process", and "animal organ development". The full datasets are represented by the bar charts of combined Biological Process and Molecular Function "most specific" enriched GO terms in Fig. S6.

To further resolve the enrichment and DEG data, separate lists of up-regulated (L2FC >1.5) and down-regulated (L2FC <−1.5) genes (same sets as used above for Fisher's Exact Test) were combined for each of the egg and sperm, indexed with their KO terms retrieved from Kofamscan (*Aramaki et al., 2019*), and then uploaded into the KEGG "Reconstruct Pathway" tool (*Kanehisa, 2017*). For every pathway, the up- and down-regulated genes were noted and based on trends seen across egg and sperm data in the above enrichment analysis, connections between KEGG pathway activity and known physiological mechanisms of egg and sperm data were made. One major finding from pathway analysis is that nearly every ribosomal protein in both the egg and sperm datasets is down-regulated (Fig. S7). Another intriguing result, when considering DEGs from the egg vs. adult tissue is that "lipid metabolic process" is among the most down-regulated GO terms. Eggs are rich in lipids and lipid activity (*Arai et al., 1993*; *Figueiredo et al., 2012*), therefore our data appeared to be anomalous, or resulting from annotation bias. However, upon further inspection, we find that nearly every down-regulated gene associated with this GO term is annotated as a lipase. Therefore, activities associated with lipid breakdown are those down-regulated in the egg.

### Sperm vs. egg comparison

PCA analysis of egg vs. sperm cDNA libraries does not show as marked a difference, as found between the egg and sperm libraries compared to the adult (PC1 of egg vs. adult shows 95% variance, PC1 of sperm vs. adult shows 89% variance, and PC1 of egg vs. sperm shows 47% variance; Fig. S4). Specifically, with respect to the DEG analysis of

egg vs. sperm RNA-seq data, these results are more limited in scope. DESeq2 identified 247 transcripts that were significantly differentially expressed (defined in this study as FDR-adjusted $p$-value < 0.05), of which 108 were up-regulated in the egg (down-regulated in the sperm) and 137 were down-regulated in the egg (up-regulated in the sperm). L2FC was >1.5, <−1.5 for all transcripts, therefore no further filtering was needed. L2FC values, all FDR-adjusted $p$-values, putative annotations, blastx hits and percent identity (PID), KO terms, and GO terms are provided for all of these genes in Table S13. The sets of egg vs. sperm DEGs were too small to do Fisher's Exact Test for GO enrichment and too sparse to fill in KEGG pathways, therefore the most useful annotations for these genes are from BLAST. The top ten DEGs that had BLAST hits for this analysis are shown in Table 2 and all annotations are listed in Table S13.

## DISCUSSION

On the most general level, egg and sperm of *M. capitata* each differ considerably in their gene expression when compared with RNA-seq data from adult cells but differ much less when compared to each other. The *M. capitata* genome contains 63,227 genes, of which 20,220 (31.98%) in the egg and 19,140 (30.27%) in the sperm are expressed, compared to 37,105 (58.69%) in the adult (these are raw counts in the RNA-seq data, prior to TPM normalization). Because gametes are specialized cells with the purpose of uniting and producing an embryo, it is not surprising that their transcriptomes are more streamlined and specialized. This is seen in our data in the down regulation of ribosomal proteins as well as in the raw RNA-seq data via TPM distributions. What is surprising, however, is the degree of similarity between egg and sperm functional capacity. In addition to their mutual down-regulation of transcription and translation, the most highly expressed genes in the egg and sperm datasets share the same core functions at similar expression levels (Table 1 and Fig. 2). Furthermore, when doing a direct comparison of egg and sperm function with differential expression analysis, DEGs were sparse (247 [most lack annotations]) compared to the numbers of DEGs found in the egg vs. adult and sperm vs. adult datasets (13,890 and 9,717 respectively) demonstrating that the degree to which the two gametes differ with respect to function is small.

### Translation

One of the starkest differences to emerge from the DEG analysis is the marked down-regulation of nearly every ribosomal protein-encoding gene in both the egg and sperm datasets when compared with the adult (Fig. S7). This trend has been documented in human and mammalian systems where sperm are "translationally silent"; i.e., cytoplasmic ribosomal assembly and thus activity is not fully functional when the sperm reaches maturity and nuclear-encoded transcripts may be primarily translated on mitochondrial ribosomes (*Gur & Breitbart, 2006*; *Zhao et al., 2009*; *Amaral et al., 2014*). This trend has also been documented in the non-coral egg literature as well with studies of *Xenopus* eggs (*Smits et al., 2014*) and in mice oocytes, where ribosomal protein expression is repressed during late stage oocyte development (*Taylor & Pikó, 1992*). As highly specialized cells with much lower overall gene expression compared to adult cells, it is not surprising for

**Table 2 Annotation of top DEGs in egg and sperm comparisons.** Annotation of the top ten DEGs with BLAST hits in the comparison of egg and sperm RNA-seq libraries. PID is protein identity.

| Gene name | Log2 fold change | *p*-adjusted | BLASTx Annotation | PID |
|---|---|---|---|---|
| *Up-regulated in egg/down-regulated in sperm* | | | | |
| adi2mcaRNA25316_R8 | 7.71554159 | 2.15E−11 | >XP_022808494.1 histone H2B-like [*Stylophora pistillata*] | 92.3 |
| adi2mcaRNA9523_R7 | 7.69856143 | 1.36E−05 | >PFX24216.1 E3 ubiquitin-protein ligase TRIM71 [*Stylophora pistillata*] | 49.2 |
| g21152 | 7.4243888 | 3.69E−05 | >XP_015772650.1 PREDICTED: tyrosine-protein kinase Fer-like [*Acropora digitifera*] | 79.4 |
| adi2mcaRNA9523_R9 | 7.25684272 | 5.25E−05 | >PFX24216.1 E3 ubiquitin-protein ligase TRIM71 [*Stylophora pistillata*] | 49.5 |
| g12388 | 7.18366792 | 7.45E−05 | >XP_020610552.1 protein Mpv17-like [*Orbicella faveolata*] | 85.9 |
| g6237 | 6.85148779 | 3.02E−04 | >PFX17025.1 Poly [ADP-ribose] polymerase 14 [*Stylophora pistillata*] | 54.3 |
| adi2mcaRNA18203_R4 | 6.40487831 | 2.14E−03 | >PFX19948.1 Coiled-coil domain-containing protein 150 [*Stylophora pistillata*] | 70.8 |
| g48873 | 6.2654406 | 6.74E−03 | >XP_020623935.1 centromere protein N-like isoform X1 [*Orbicella faveolata*] | 61.6 |
| g18750 | 6.17777817 | 4.36E−03 | >PFX27818.1 Amiloride-sensitive amine oxidase [copper-containing] [*Stylophora pistillata*] | 63.2 |
| g28939 | 5.94022377 | 2.58E−03 | >XP_020610555.1 exostosin-3-like [*Orbicella faveolata*] | 72.7 |
| *Up-regulated in sperm/down-regulated in egg* | | | | |
| g9290 | 11.493954 | 1.24E−03 | >XP_015776821.1 PREDICTED: creatine kinase B-type-like isoform X1 [*Acropora digitifera*] | 83.9 |
| g34129 | 9.4025289 | 1.23E−06 | >XP_015757707.1 PREDICTED: fibropellin-1-like isoform X3 [*Acropora digitifera*] | 50.5 |
| g22489 | 8.8676566 | 3.42E−06 | >XP_015774900.1 PREDICTED: cofilin-like isoform X1 [*Acropora digitifera*] | 79.2 |
| g24088 | 8.4749941 | 7.84E−11 | >XP_020610152.1 ADP-ATP carrier protein 3, mitochondrial-like [*Orbicella faveolata*] | 87.4 |
| g43213 | 8.228315 | 4.44E−05 | >XP_020620813.1 UPF0573 protein C2orf70 homolog B-like [*Orbicella faveolata*] | 63.8 |
| g1536 | 8.0816351 | 4.44E−05 | >XP_020600610.1 lymphocyte antigen 6H-like isoform X2 [*Orbicella faveolata*] | 48.5 |
| g50437 | 7.8691611 | 4.02E−04 | >XP_015777745.1 PREDICTED: agrin-like [*Acropora digitifera*] | 50.5 |
| g24451 | 7.7600705 | 7.53E−05 | >XP_020615035.1 MORN repeat-containing protein 5-like [*Orbicella faveolata*] | 55.8 |
| g61108 | 7.5970964 | 3.11E−03 | >XP_015777385.1 PREDICTED: stabilizer of axonemal microtubules 2-like isoform X2 [*Acropora digitifera*] | 80.4 |
| g48486 | 7.5070947 | 5.19E−03 | >XP_015756818.1 PREDICTED: mitochondrial glutamate carrier 2-like [*Acropora digitifera*] | 86.2 |

ribosomal protein genes and thus, translation to be down-regulated in both the sperm and egg. However, this topic needs to be further studied to determine whether this phenomenon in sperm is due solely to its role as a gamete or because the cytoplasmic ribosomes may be translationally inactive.

## Fertilization

In broadcast-spawning corals, there are pre-zygotic barriers that gametes must overcome to achieve fertilization (*Monteiro, Serrão & Pearson, 2012*; *Monteiro et al., 2016*), including successful chemical signaling between gametes. Chemical signals are secreted by egg cells to attract sperm. Whether this process is utilized to guide sperm to an egg within the body of an individual organism or the broader environment depends on the system being studied. Regardless, there are two major components to this process: secretion of the chemical attractant by the egg, and taxis initiated by chemical receptors, and made physically possible by motile cilia/ the flagellum of the sperm. In mammalian sperm, for example, motile cilia and the sperm flagellum develop in a similar fashion, have the same axoneme structure, and are virtually identical (*Clermont, Oko & Hermo, 1993*; *Avidor-Reiss & Leroux, 2015*; *Wachten, Jikeli & Kaupp, 2017*). The evolution of compartmentalized sperm cilia (via primary cytosolic ciliogenesis) is necessary for the flagellar movement that propels the sperm cell through its environment to the egg. This is thought to be a universal process in metazoans because it is present in both protostomes (*Drosophila*) and deuterostomes (mammals, humans), as well as in basal metazoans (corals, sponges) (*Avidor-Reiss & Leroux, 2015*). In broadcast-spawning corals like *M. digitata* (a relative of *M. capitata*), eggs play a significant role in the regulation of sperm activity and fertilization success, including sperm signaling and stimulation of flagellar motility (*Coll et al., 1994*; *Morita et al., 2006*). The signals produced by the egg must be species-specific to prevent hybridization during mass spawning events where many species release gametes into the water column at the same time. In organisms like ascidians and echinoids, $Ca^{2+}$ has been shown to induce sperm flagellar motility (*Yoshida et al., 2003*; *Morita et al., 2006*). A similar effect has been shown experimentally in corals in *Acropora* species although this alone is insufficient to explain the species-specific nature of sperm and egg union (*Morita et al., 2006*). It is difficult to find evidence of these complex processes in the summary data of *M. capitata* discussed above due to the large number of genes lacking annotations. However, upon taking a closer look at the GO data used for Fig. 2, for "motile cilium", 79 genes are associated with this GO term, 38 are present in the sperm with TPM counts >100, and two of those (g63277 and g9762) are also linked to the GO term "calcium ion binding". Neither of these genes were assigned a KEGG annotation which is why the data for flagellar and motile cilia cellular components are scarce for this analysis. Both of these genes have the same top blastx hit: XP_015777656.1 PREDICTED: uncharacterized protein LOC107355583 isoform X1 in *Acropora digitifera*. Both of these genes have transcripts that are significantly up-regulated in the sperm vs. adult data and egg vs. adult data. Based on these results, it is possible these genes play a key role in communication associated with the fertilization process. Differential expression and BLAST annotation statistics for these two genes are presented in Fig. S8. We speculate that these genes are interesting targets for future CRISPR/Cas9 based gene knockdowns (recently developed for Scleractinia; *Cleves et al., 2018*; *Cleves et al., 2020*) to explore coral reproductive biology.

### Differential expression between egg and sperm

As mentioned throughout this study, it is difficult to gain conclusive insights into the functional capabilities of different *M. capitata* cell types due to the lack of annotations for many DEGs. Of the 108 genes significantly up-regulated in the egg when compared to sperm, 26 of these DEGs (∼24%) have top blastx hits to uncharacterized proteins and 12 (∼11%) lack an annotation. Of the 139 significantly up-regulated genes in sperm, 32 (∼23%) have top blastx hits to uncharacterized proteins, whereas 13 (∼9%) lack an annotation. These results demonstrate that ca. 1/3 of the genes in this dataset, including many of the most highly differentially expressed genes (based on L2FC), do not provide functional information. It is worth noting however, that many of the annotated DEGs in the egg set encode proteins that are localized to the nucleus and take part in mitosis-related or DNA-based processes, whereas the proteins encoded by sperm DEGs often have functions related to motility and ATP-binding. These limited findings are consistent with general gamete physiology, whereby eggs express cell division, transcription, and DNA repair functions and sperm are optimized for energy production to support motility.

## CONCLUSIONS

Our findings suggest that coral egg and sperm are not highly differentiated with respect to functional capability. Rather, as motile cells released into the water column and subjected to the same environment prior to fertilization, these morphologically divergent cell types share a conserved gene expression pattern and thus, may be under similar functional constraints. To this end, it would be of interest to investigate egg and sperm transcriptomics in brooding corals (i.e., species that do not release gametes into the water column prior to fertilization) and compare these data to the results of our study.

In addition to the major finding of largely shared expression profiles, the data also provide insights into which genes may play key roles in the fertilization process. Due to an association with cell motility and calcium ion binding, two genes (g63277 and g9762) emerge from our dataset as possible candidates for future experimental studies on how eggs and sperm recognize each other in the water column despite the presence of many other gametes from other organisms. We recognize however that our findings are largely summary in nature and serve as an initial step in understanding the transcriptome (vis-à-vis functional capacity) of *M. capitata* gametes. In conclusion we present here the first study of the transcriptome of coral sperm and eggs and reach interesting conclusions that pave the way for future multi-omics and genetics investigations on this topic (*Cleves et al., 2020*), particularly in the context of anthropogenic climate change influences on the marine environment.

## ACKNOWLEDGEMENTS

We would like to acknowledge Prof. JunMo Lee and Ehud ZelZion for their guidance with various bioinformatics programs.

### Funding

This work was supported by grants from the US National Science Foundation (NSF OCE IEP 1756616 to Debashish Bhattacharya and NSF OCE, IEP, EPSCoR 1756623 to HMP), the Israeli Binational Science Foundation (BSF 2016321 to Hollie Putnam and Tali Mass), and the Paul G. Allen Philanthropies. Debashish Bhattacharya was supported by a NIFA-USDA Hatch grant (NJ01170). The funders had no role in study design, data collection and analysis, decision to publish, or preparation of the manuscript.

### Grant Disclosures

The following grant information was disclosed by the authors:
US National Science Foundation: NSF OCE IEP 1756616, NSF OCE, IEP, EPSCoR 1756623.
Israeli Binational Science Foundation: BSF 2016321.
NIFA-USDA Hatch: NJ01170.

### Competing Interests

The authors declare there are no competing interests.

### Author Contributions

- Julia Van Etten conceived and designed the experiments, analyzed the data, prepared figures and/or tables, authored or reviewed drafts of the paper, and approved the final draft.
- Alexander Shumaker analyzed the data, prepared figures and/or tables, and approved the final draft.
- Tali Mass performed the experiments, authored or reviewed drafts of the paper, and approved the final draft.
- Hollie M. Putnam conceived and designed the experiments, performed the experiments, authored or reviewed drafts of the paper, and approved the final draft.
- Debashish Bhattacharya conceived and designed the experiments, authored or reviewed drafts of the paper, and approved the final draft.

### Ethics

The following information was supplied relating to ethical approvals (i.e., approving body and any reference numbers):

The sperm and egg collection in Hawaii was authorized by the Hawaii Institution of Marine Biology Special Activity Permit 2018-50.

### DNA Deposition

The following information was supplied regarding the deposition of DNA sequences:

The coral sperm genomic and transcriptomic data created for this project are available at NCBI BioProject PRJNA339779.

The three libraries were pooled together in BioSample: SAMN05607941; Sample name: Mcap_nonBleach_RNA; SRA: SRS1632867. The egg transcriptomic data are available at

NCBI BioProject PRJNA616341 and the IDs of the three libraries are SAMN14486762: 119_egg_AMB (TaxID: 46704), SAMN14486763: 120_egg_AMB (TaxID: 46704), and SAMN14486764: 121_egg_AMB (TaxID: 46704).

## Data Availability

Code for RNA-seq data processing and differential gene expression in RStudio are available in the Supplemental Files.

## Supplemental Information

Supplemental information for this article can be found online at http://dx.doi.org/10.7717/peerj.9739#supplemental-information.

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
