# Peer review of "Transcriptome analysis provides a blueprint of coral egg and sperm functions"

_PeerJ, doi:10.7717/peerj.9739_

## Round 0.1 · original submission · Major Revisions

Two expert reviewers have evaluated your resubmission and their comments can be seen below. One reviewer in particular has some important suggestions and comments, which need to be attended to.

·

Basic reporting

This is a timely and important study on gene expression in gametes of a reef building coral. The methodology is sound as are the analyses. The findings are interesting on their own, but have value in supporting much-needed research on molecular responses of different life-history stages of corals to environmental stressors.

Experimental design

The protocols are all appropriate to addressing the questions posed. The collection and handling of gametes were properly performed. The analytical techniques are sound as are the data analyzes.

Validity of the findings

At a time when coral reef throughout the world are in steep decline, understanding the mechanisms underlying documented problems, including in gamete quality, fertilization success and larval recruitment, is very important. I like the use of the KEGG and GO databases in the analyses, as these provide important insight. This is a well-designed and executed study.

Additional comments

My editorial suggestions are included in the manuscript for their consideration.

Reviewer 2 ·

Basic reporting

no comment

Experimental design

These are interesting data sets and should certainly be published in some form. However, I feel the authors have not completely found their story yet, and some analytical choices seem peculiar given the stated hypotheses. I looked at the authors’ rebuttal letter to the previous round of reviews and was not surprised to find that the scope and analyses had changed quite a bit from the original intent. That comes across in the manuscript in a few ways. My greatest concern is that some of the experimental design choices undermine the validity of the few conclusions that are made.

Major Concerns

1. There are so many sources of gene expression variation that weren’t accounted for. The sperm, adult, and egg samples were collected across different years (2015, 2016, and 2018, respectively), collected from different sets of genetically distinct colonies and then pooled from different numbers of colonies (I could be wrong here—the methods are quite vague regarding collection), processed very differently (e.g. some with Qiagen kits, others with Zymo kits), and run on different high-throughput sequencers with different sensitivities (MiSeq produced 30 million reads for sperm while HiSeq produced 125 million reads for eggs). I realize that pooling and genetic variation might not be a big deal in gamete samples, which are by their nature composed of millions of genetically distinct cells, and that normalizing to TPM should account for read depth differences. But when you think about how many sources of variation there are, it’s all the more surprising that eggs and sperm were found to express pretty much all the same gene pathways. I worry that this might be a bias stemming from 1) high variation within and among samples washing out the signal and 2) the somewhat arbitrary 100 TPM cutoff for detecting “expressed” genes. Is it possible that only a subset of the most highly expressed house-keeping genes were retained after the filtering step, whereas many other important, variable, and somewhat more lowly-expressed genes were missed? I don’t know the answer—perhaps some sort of rarefaction analysis could be performed. I know its not that useful to say I have vague concerns about the validity of the results, but as this is the main conclusion of the study, I’m giving it more scrutiny and ultimately I think it needs more support.

2. The first line of the discussion reads “On the most general level, the egg and sperm of M. capitata each differ considerably in their gene expression when compared with RNA-seq data from adult cells but differ much less when compared with each other.” To me, that implies that three DEG analyses were performed: egg vs. adult, sperm vs. adult, and most importantly egg vs. sperm. But as far as I can tell this was not the case; there was no direct egg vs sperm comparison, and it really begs the question as to why. Instead the authors looked at overlap in DEGs from the sperm vs. adult contrast and the egg vs. adult contrast, but this set does not necessarily capture all of the genes that could have been differentially expressed between sperm vs. egg directly. For example, Gene A could be 2-fold upregulated in sperm relative to adults and 2,000-fold upregulated in eggs vs adults. Since they are both differentially upregulated relative to adults, they would both fall in the “shared” portion of the Venn diagram of Figure 2A. But if they were compared directly it would be obvious that their expression patterns were quite distinct, and lumping them as sharing an expression pattern would be an oversimplification. Again, eggs and sperm sharing patterns is pretty much the major conclusion of the paper, but it wasn’t tested directly. The authors either need to run that test or provide a very clear explanation as to why it doesn’t need to be done.

3. A full half of the discussion is devoted to speculation about the role of two genes in fertilization. It’s too much—I’d reduce this section down to two or three sentences, and perhaps move some of the material up to the introduction. Overall I think this is a minor finding and doesn’t have very much solid evidence to support it. If you make this change, there’s really very little discussion left. It’s an unbalanced paper in that respect, which is one of the reasons I don’t think the story has been fleshed out very well.

Validity of the findings

Please see "Experimental Design" section, since I covered "Validity" aspects in most comments.

Additional comments

Minor Concerns

4. I really don’t understand the value of Figure 2. It is meant to show similarities and differences in sperm/egg expression relative to adults, but it is restricted to only the top 10 most highly upregulated or downregulated genes, which may not be all that representative. I would rather see counts of the total number of upregulated or downregulated genes that are distinct or shared across sperm/eggs relative to adults. That would give a much more representative summary. I think one of the goals was to show that many of the top genes of interest weren’t annotated, but you don’t need a figure to get that point across.

5. Figure 1 looks very nice but I find it hard to interpret. The text suggests order matters but the caption doesn't provide details. Originally I thought a heatmap would be better but the text indicates there are no major differences. So what exactly is being highlighted? If it's just a list of shared expressed pathways, wouldn't a pathway map be a better visual representation?

6. In the methods, can you provide any details about the sperm/egg concentrations in the final samples? If they were maintained for a half hour in 1.5 mL tubes at very high concentrations, I worry about anoxia kicking in, causing stress, and potentially harmonizing expression among sperm and eggs. Although if that happened, I imagine we’d see more enrichment of stress-related genes. I’d also appreciate a clearer explanation for how sample pooling happened, and whether the three replicates were per sample, or only per treatment.

7. L295: Are the “egg up-regulated genes” inclusive or exclusive of the sperm up-regulated genes that overlap? Same with the down-regulated genes.

8. L296: I’d keep LF2C capitalized throughout the manuscript to be consistent.

9. L324-325: I’m having a hard time wrapping my head around whether the sperm/egg percentages of the exome are comparable to the adult percentage. Which TPM cutoff was used for the adult data set in this case, 100 or 60?

10. L332-334: Need to make clear that the DEG lists in Figure 2 are similar in identity but not necessarily similar in expression level (all you can infer is a similar direction). This is why I think it’s a bit disingenuous to say that sperm and egg have similar expression without testing it directly.

---

## Round 0.2 · accepted · Accept

I am satisfied with the changes made to the manuscript.

Reviewer 2 ·

Basic reporting

no comment

Experimental design

no comment

Validity of the findings

no comment

Additional comments

The authors did an excellent job addressing my concerns. I was very glad to see that the new DEG results were consistent with the initial conclusions. With that extra level of support, I found the revised manuscript's story to be more cohesive and convincing than the original. Great work.